# The Infancia y Procesamiento Sensorial (InProS—Childhood and Sensory Processing) Project: Study Protocol for a Cross-Sectional Analysis of Parental and Children’s Sociodemographic and Lifestyle Features and Children’s Sensory Processing

**DOI:** 10.3390/ijerph17041447

**Published:** 2020-02-24

**Authors:** Paula Fernández-Pires, Desirée Valera-Gran, Alicia Sánchez-Pérez, Miriam Hurtado-Pomares, Paula Peral-Gómez, Cristina Espinosa-Sempere, Iris Juárez-Leal, Eva-María Navarrete-Muñoz

**Affiliations:** 1Department of Surgery and Pathology, Miguel Hernández University, 03550 Alicante, Spain; paula.fernandezp@umh.es (P.F.-P.); alicia.sanchez@umh.es (A.S.-P.); mhurtado@umh.es (M.H.-P.); pperal@umh.es (P.P.-G.); c.espinosa@umh.es (C.E.-S.); ijuarez@umh.es (I.J.-L.); enavarrete@umh.es (E.-M.N.-M.); 2Grupo de Investigación en Terapia Ocupacional (InTeO), Miguel Hernández University, 03550 Alicante, Spain

**Keywords:** sensory processing, childhood, development, health, lifestyle

## Abstract

Sensory processing difficulties may have potential detrimental consequences on the physical, social and intellectual development of children. It includes serious disturbances affecting emotional regulation, motor performance, social behavior and daily life functioning, among others factors. Since these symptoms are more frequent among children with disabilities, most research has been carried out in clinical populations. However, recent studies have suggested that sensory problems may be prevalent in around 20% of children without clinical conditions. To date, epidemiological research on sensory dysfunctions in normally developing children is lacking; therefore, it is unknown whether or not sensory processing difficulties are significant factors that affect child’s development. Hence, this study has a double general purpose: (1) to determine the sensory profile of school-aged children; (2) to examine the associations between atypical sensory processing patterns and socio-demographic, health and lifestyle features of these children and their parents. The Infancia y Procesamiento Sensorial (InProS, Childhood and Sensory Processing in English) project is a population-based cross-sectional study of Spanish children aged 3–7 years. Data were gathered from different ad hoc questionnaires and several standardized tests. We propose an objective and reliable methodology using statistical and research procedures to describe and determine associations with sensory processing outcomes. We believe that this project will contribute to filling the gap in epidemiological research on sensory issues by providing more convincing evidence. Nevertheless, the potential results should be corroborated in other larger samples.

## 1. Introduction

Sensory processing is a complex neurological ability that integrates information received from our sensory systems (i.e., tactile, olfactory, gustatory, visual, auditory, proprioceptive and vestibular). As a part of the process, sensory information is interpreted, selected and organized by our brain to produce adequate motor, behavioral, emotional and/or attentional responses [1]. The presence of difficulties in sensory processing can manifest as impaired responses to, processing of, and/or organization of sensory information that may compromise the normal development of children by affecting their participation in functional daily life routines and activities [2,3]. Importantly, sensory processing dysfunction may affect the physical, social and intellectual development of children including problems related to emotional regulation, motor performance, social interaction, and daily life functioning at home, at school, and in the community [4].

Atypical sensory processing is a common feature in children with developmental problems such autism spectrum disorder (ASD) or attention deficit hyperactivity disorder (ADHD) [3], although current research suggests that 5% to 18% of children from the general population aged between 3 and 11 years have symptoms associated with sensory processing difficulties [5,6,7,8]. However, the existing literature to date on sensory processing issues is still limited and heterogeneous [4]. In this respect, it should be noted that children diagnosed with certain clinical conditions are more likely to present sensory processing challenges, which explains that most research has been mainly conducted in clinical populations. Moreover, closely linked to this limitation, it should be considered that the available evidence has essentially emerged from isolated findings due to the samples studied have diverse features and comorbidities. Nevertheless, certain improvement in the understanding about sensory processing patterns in various clinical populations has been observed recently although, at the same time, it has been also recognized that there is an important lack of knowledge with respect to children without clinical diagnoses [4].

As far as we know, epidemiological research about sensory processing difficulties in normally developing children has basically yielded prevalence data provided from scarce studies [9,10]. Indeed, there are no previous studies having explored factors associated with sensory processing dysfunction that could affect the normal development of children without disabilities. In this regard, although establishing prevalence rates of sensory processing difficulties is an essential step to formulate etiological hypothesis, to provide more convincing evidence, it is important to use other epidemiological approaches to investigate the exposure/outcome associations of interest. Consequently, at this stage, more efforts should be made to ensure new good quality epidemiological studies in sensory processing difficulties in children from different countries and populations.

Therefore, this study attempts to fill the gap in epidemiological research on sensory processing issues in typically developing children. Firstly, this study aims to determine the sensory profile of school-aged children. Secondly, based on prior knowledge about socio-demographic and lifestyle factors related to child neurodevelopmental outcomes from solid evidence [11,12,13,14,15], we examine the associations between the detected sensory processing patterns and socio-demographic, health and lifestyle characteristics of the parents and their children. To the best of our knowledge, this is the first observational population-based study aimed to explore determinants affecting sensory processing in typically developing children in Spain. We think that an early detection of these problems could positively contribute to determine whether or not sensory processing difficulties are significant factors that affect child’s development. Hence, this research will yield useful information to design appropriate interventions for specific sensory processing and associated functional difficulties in order to prevent detrimental consequences in child’s health and in later life.

## 2. Materials and Methods

### 2.1. Study Design and Participants

The InProS project is a population-based cross-sectional study of children aged 3–7 years living in the province of Alicante (population > 1,830,000 inhabitants) in Spain. Participants were selected from a random sample of 21 schools registered at the office of the Consellería de Educación, Cultura y Deporte de la Generalitat Valenciana (Education, Culture and Sport Council of the Provincial Government, http://www.ceice.gva.es). More details about the study are available at www.inteo.edu.umh.es/inpros/.

### 2.2. Procedure and Enrolment

The enrolment was performed from February to May 2016. Once the permission from the principal of each school was given, around 1700 eligible children were invited to participate in this study through an envelope with an invitation letter addressed to their parents. The envelope contained a participant information sheet with the main project details, a booklet with several questionnaires, as well as the instructions on how to complete them. After approximately a 2/3-week period, all children were asked to return the completed questionnaires, along with the informed written parental consent. Once all the documentation was examined, children were excluded from the study if they presented any disability. A total sample of 620 children was finally included, rendering a response rate of approximately 37%. 

### 2.3. Study Variables

Information about sociodemographics, health and lifestyle behaviors of the children and their parents was collected using different ad hoc questionnaires and several measurements based on previous studies. A summary of the data collection undertaken in the InProS study is presented in Table 1.

#### 2.3.1. Main Outcome Measure: Child Sensory Processing

Data on the child sensory processing was measured by the cross-culturally adapted and validated version of the short sensory profile (SSP) for Spanish children (SSP-S) [20,21]. The SSP is a screening tool based on the Sensory Profile, a questionnaire designed by W. Dunn (1999) [19] and used to identify sensory processing difficulties. It is a parent report measure that consists of a 38-item questionnaire divided into seven sections or subscales that collect information on different sensory issues: tactile sensitivity, taste/smell sensitivity, movement sensitivity, under-responsive/seeks sensation, auditory filtering, low energy/weak, and visual/auditory sensitivity. All items are scored on a one-point to five-point scale (i.e., ranging from 1–always to 5–never). As displayed in Table 2, the SSP total score and the score on each subscale can be obtained by summing up the respective values of the items. Moreover, the yielded scores can be used to determine children’s sensory profile (typical performance, probable difference, or definite difference) according to the cut-points proposed by Dunn [19]. 

#### 2.3.2. Other Outcome Measures: Sociodemographic Features, Lifestyle Behaviors and Health Conditions

Information about child and parental characteristics of interest for the study were gathered from different ad hoc questionnaires and several standardized tests that were reported by the parents. The selection of the tests and the design of the ad hoc questionnaires were based on previous studies focused on socio-demographic and lifestyle factors related to child neurodevelopmental outcomes [11,12,13,14,15], as well as on the experience learned from the participation by the research team in mother-child studies such as the INMA study [22]. To formulate the questions about child’s feeding problems and oral-facial problems, we counted on several occupational therapists trained in sensory integration with broad clinical experience. For more details, the booklet with the questionnaires used in this study is available at: http://inteo.edu.umh.es/wp-content/uploads/sites/1447/2020/01/CUESTIONARIO-final1.pdf.

### 2.4. Data Management

All data obtained from questionnaires was entered in Microsoft Office Excel (Microsoft Corporation, Redmond, WA, USA) spreadsheets to create a database. Each participant was given a unique identification number to protect the confidentiality of the personal information. The principal investigator designated a research assistant as responsible for ensuring the appropriate data management and storage. Electronic files were backed up, maintained, and stored on a hard drive, and hard copy original material including questionnaires, tests and personal data were stored in numerical order in binders in a secure cabinet. Access to the study data will remain restricted, and all the files will be maintained in storage for a minimum period of ten years after completion of the study.

### 2.5. Statistical Analysis

#### 2.5.1. Sample Size

Although existing literature has indicated that the prevalence of sensory problems in children without disabilities can reach up to 55% [23], it is generally assumed that rates can range between 10% and 20% in children from the general population [24]. Thus, to calculate the sample size the following assumptions were used: a prevalence of 18%, a significance level of 5%, a power of 80% and a two-sided test, thereby obtaining a sample of 485 participants as optimal. Moreover, to compensate the lack of statistical power because of the likely nonresponse, a rate of 15% of nonresponse was assumed according to the following formula: 485×100÷85. Therefore, a final sample of 570 would be required to meet the sample size requirements of the study. All procedures carried out for the estimation of the sample size were performed using software R, version 3.6.1 (R Core Team. R: A language and environment for statistical computing. R Foundation for Statistical Computing (Vienna, Austria; http://www.r-project.org).

#### 2.5.2. Data Analysis Plan

Statistical analyses will be conducted using software R, version 3.6.1. All statistical tests were bilateral assuming a significance level of 5%. Descriptive analyses will be estimated to describe the sensory profile of children and sociodemographic, health and lifestyle characteristics of them and their parents. Values will be expressed as mean and standard deviation for normally distributed continuous variables or as median and interquartile range for non-normally distributed continuous variables. Kolmogorov-Smirnov test will be used to check the normal distribution of the continuous variables. Categorical variables will be displayed as frequencies and percentages.

To explore the differences in the study variables between those children classified as having a typical sensory profile (i.e., ≥155 points) and those with an atypical sensory profile (i.e., <155 points), we will apply Chi-square or Fisher’s exact for categorical variables, and T-student test or U de Mann Whitney test for continuous variables.

Bivariate regression models will be used to assess the relationship between SSP scores and study covariates and to build core models with the identified potential confounders using all the significant covariates (*p* < 0.20). Following a backward elimination procedure, all the covariates associated with the SSP scores will be included at a level of *p* <0.10. Regardless of their statistical significance, the previous variables will be kept in the models if they changed the magnitude of the main effects by more than 10%.

The association between the sensory profile (i.e., typical vs atypical performance), using the SSP total score and the score of each subscale, and the study covariates of interest will be analyzed by multiple Poisson regression models with robust variance based on the Huber sandwich estimate to obtain prevalence ratios (PR) and their 95% confidence interval (CI) [25,26]. Finally, sensitivity analyses will be also conducted to evaluate the robustness of the main findings. This includes making a set of assumptions on the potential role of the parental and child conditions in child’s sensory performance. Moreover, we will explore the differences among those children with atypical sensory profile (i.e., children classified as having a sensory profile with probable difference and those with definitive difference).

### 2.6. Ethical Aproval, Ethical Considerations and Dissemination

This study protocol obtained the ethical approval from the Research Compliance Office of the Miguel Hernández University (DPC.ASP.02.16). The research was performed in accordance with the Declaration of Helsinki.

All the participant schools were informed vocally and in writing about the project. The children and their parents received general information about the project in writing. The details of the project indicated that their participation in the study was voluntary. All participants provided informed consent and had no incentive to take part in this study. Data confidentiality is warranted during the whole research process (i.e., data collection, data cleaning and dissemination of research results). Schools, parents and children will be informed on the progress of the study. Findings from this study will be presented at international meetings and will be published in open access peer reviewed journals. Furthermore, this study will be the subject of a doctoral thesis (P.F.-P.) and will meet the requirements the PhD program at the Miguel Hernández University (Alicante, Spain).

## 3. Discussion

This cross-sectional study has been designed to determine sensory processing difficulties and explore the associated factors in typically developing children. On the basis of the available data, this research proposal has been developed in response to the lack of knowledge of sensory processing issues in children from the general population as well as to the emerging concerns arise from notable prevalence rates detected in children without clinical conditions. By specifically adopting an epidemiological approach, we hope to provide more convincing evidence about sensory processing problems that may affect detrimentally the normal development and health of school-aged children without disabilities.

Research on sensory processing issues in children is relatively recent and, to a large extent, concentrated on examining clinical populations. Since sensory processing problems are estimated at 40%−88% in children with disabilities [23], most previous studies have mainly focused on characterizing these symptoms in children affected by developmental problems such as ASD or ADHD. Simultaneously, some studies have suggested that between 10% and 55% of normally developing children may exhibit signs of sensory processing challenges [23], although the information on this population gained from research is still clearly lacking. Thus, considering that sensory processing difficulties may have a detrimental effect on the normal development of children without disabilities, the present study intends to cover this important aspect of child’s health.

One significant limitation of the current data on sensory processing issues is that the absence of studies using an epidemiological approach. This approach can offer several advantages in order to achieve effective research [27]. Firstly, epidemiological studies are population-based studies in contrast to clinical studies that usually examine small numbers of participants [28]. This population perspective allows the study findings can be interpreted in terms of group at risk (e.g., children) within particular socio-demographical context, thereby ensuring data can be extrapolated to a larger population with similar features. Secondly, epidemiological studies can provide an insight into the determinants and health-related outcomes or events that could be linked to the health problem of interest (i.e., sensory processing problems) using statistical and methodological procedures to describe, explain and predict these health events. Finally, since epidemiological studies are community-based studies, the insights gained from research should be applied to propose practical public health strategies and to tailor appropriate interventions in order to monitor and prevent health-related problems in the community [27].

This study has several limitations that should be acknowledged. The cross-sectional design of our study does not permit us to establish a temporal relationship between covariates and sensory processing outcomes. Thus, we will not be able to infer causation from all the potential associations that could be found but they will be useful to formulate further research hypothesis that are crucial for the advancement of knowledge. Although the effect unknown factors, residual confounding, or bias due to information not collected cannot be ignored, a wide range of potential confounding factors were considered to perform the main analyses. Moreover, all data were self-reported, suggesting that some misclassification cannot be disregarded, although any inaccuracy in reporting should be considered as non-differential. In this respect, it should be noted that all the questionnaires used to gather information from the study participants were valid and reliable instruments used in previous research. Another limitation can be attributed to the fact that sensory processing dysfunction has been established using the cut-off points estimated in children from the United States. However, as showed for Spanish population aged 11 and older [7], it is expected that the normative values for Spanish children at school age are similar to those in the United States.

This study also presents several notable strengths. This study collected a sample of school-aged children from the general population that was randomly selected in order to preserve the representativeness of data. This project will allow not only the detailed description of sensory profile in representative sample of Spanish children, but also the exploration of different associations between child and parental sociodemographic, health and lifestyle features and child sensory processing outcomes. We believe that the research methodology and tools we propose will yield accurate and use information to provide more convincing evidence. Nevertheless, we are aware that the potential results should be corroborated in other larger samples. 

## 4. Conclusions

This study provides an epidemiological approach to the study of sensory processing difficulties in school-aged children from the general population. A considerable proportion of children without disabilities may be affected by these health problems, although it is unknown whether or not sensory processing difficulties are significant factors that have a negative impact on child’s development. In this study, we propose an objective and reliable methodology using statistical and research procedures to describe and determine associations with sensory processing outcomes. In this respect, this research proposal attempts to fill the gap in epidemiological research on sensory processing in typically developing children. Thus, we hope that our findings can constitute a suitable rationale for replicating in further samples using a prospective study design.

## Figures and Tables

**Table 1 ijerph-17-01447-t001:** Summary of the data collection for the InProS study.

Outcome Meaures	Participants			Measurement Method
	Mother	Father	Child	
Sociodemographics				
Age	x	x	x	ad hoc questionnaire
Country of origin	x	x		ad hoc questionnaire
Education level	x	x		ad hoc questionnaire
Marital status	x	x		ad hoc questionnaire
Work status	x	x		ad hoc questionnaire
Lifestyle behaviors				
Sleep (h/d)	x	x	x	ad hoc questionnaire
Sleep quality			x	Pediatric Sleep questionnaire [16]
TV watching (h/d)	x	x	x	ad hoc questionnaire
Sendentary activities (h/d)			x	ad hoc questionnaire
Global physical activity	x	x	x	ad hoc questionnaire
Extracurricular Activities			x	ad hoc questionnaire
Smoking	x	x		ad hoc questionnaire
Dietary quality			x	KIDMED index [17]
**Health**				
Parental stress	x	x		Parental Stress scale [18]
Reproductive history	x			ad hoc questionnaire
Parity	x			ad hoc questionnaire
Pregnany complications	x			ad hoc questionnaire
Smoking during pregancy	x			ad hoc questionnaire
Gestation weeks	x			ad hoc questionnaire
Type of delivery	x			ad hoc questionnaire
Birth weight			x	ad hoc questionnaire
Lactancy	x		x	ad hoc questionnaire
Medical conditions			x	ad hoc questionnaire
Drug intake			x	ad hoc questionnaire
Vitamin/mineral intake			x	ad hoc questionnaire
Feeding problems			x	ad hoc questionnaire
Oral-facial problems			x	ad hoc questionnaire
Sensory processing			x	Short Sensory Profile [19]

Abbreviations: InProS, Infancia y Procesamiento Sensorial (Childhood and Sensory Processing); h, hours; d, day; TV, television.

**Table 2 ijerph-17-01447-t002:** Summary of the scores for the SSP total and total SSP subscales and classification of the child sensory profile.

SSP Section/Subscale (Items)	Classification			
	Range Score	Typical Performance	Probable Difference	Definite Difference
Tactile Sensitivity (1−7)	7−35	35−30	29−27	26−7
Taste/Smell Sensitivity (8−11)	4−20	20−15	14−12	11−4
Movement Sensitivity (12−14)	3−15	15−13	12−11	10−3
Underresponsive/Seeks Sensation (15−21)	7−35	35−27	26−24	23−7
Auditory Filtering (22−27)	6−30	30−23	22−20	19−6
Low Energy/Weak (28−33)	6−30	30−26	25−24	23−6
Visual/Auditory Sensitivity (34−38)	5−25	25−19	18−16	15−5
Total (1−38)	38−190	190−155	154−142	141−38

Abbreviations: SSP = Short Sensory Profile.

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
