# Peer review of "The Infancia y Procesamiento Sensorial (InProS—Childhood and Sensory Processing) Project: Study Protocol for a Cross-Sectional Analysis of Parental and Children’s Sociodemographic and Lifestyle Features and Children’s Sensory Processing"

_ijerph, 2020, doi:10.3390/ijerph17041447_

Round 1
Reviewer 1 Report
The article describes a protocol for a much needed cross-sectional study focusing on the association between family and children’s sociodemographic and lifestyle factors and children’s sensory processing. Protocol studies are important as they serve as a quality control tool. Therefore, this article will contribute to post-peer review the findings and conclusions of the resulting study and it is suitable for publication in IJERPH.
The paper is overall well written and organized. The authors give a nice overview of sensory processing definition and epidemiology, which will help unfamiliar readers to understand aspects regarding sensory processing. Moreover, it is really good that authors include lifestyle related variables that go often overlooked in most of sensory processing studies. Authors also describe additional valuable information that could be of help for future studies, such as the response rate, which could inform other researchers of what to expect if they are to conduct a population-based study.
The approach and methodology proposed are sound and will likely make up for any potential limitations. However, there are some aspects that should be revised.
General aspects
1.0. This is a minor typo. Line 1: it should be “protocol” instead of “article”.
1.1. Authors use several terms to refer to alterations in typical sensory processing, which may confuse readers (i.e., sensory processing difficulties, sensory processing issues, sensory processing dysfunction, sensory processing patterns). Some of these terms are outdated (i.e., sensory processing dysfunction) or misleading (sensory processing patters are better examined with the Sensory Profile-2). Consistent use of this term throughout the text will help readers to get a better idea about this condition. I recommend to adhere to the classification proposed by Miller et al. and to use “sensory processing disorders”:
Miller LJ, Anzalone ME, Lane SJ, Cermak SA, Osten ET. Concept Evolution in Sensory Integration: A Proposed Nosology for Diagnostic. Am J Occup Ther. 2007;61:135-40.
1.2. In the Introduction, authors provide a nice overview of sensory processing disorders (SPD) prevalence and epidemiology. However, data comes mostly from North American children, and prevalence of SPD may present differently according to context and country. Authors may find the following related papers interesting to include in the protocol, as they explore prevalence of SPD in different countries using the Short Sensory Profile:
Engel-Yeger B. The applicability of the short sensory profile for screening sensory processing disorders among Israeli children. Int J Rehabil Res. 2010;33:311-8.
Román-Oyola R, Reynolds S. Prevalence of Sensory Modulation Disorder among Puerto Rican Preschoolers: An Analysis Focused on Socioeconomic Status Variables. Occup Ther Int. 2013;220:144-45.
Delgado-Lobete L, Montes-Montes R, Rodríguez-Seoane S. Prevalencia de Trastorno del Procesamiento Sensorial en niños españoles. Resultados preliminares y comparación entre herramientas de diagnóstico. TOG. 2016;13:4.
1.3. Authors estimate the sample size on the assumption of a prevalence of SPD=18%. A recent preliminary study conducted in Spain (see reference above) found a SPD prevalence of 14.3%, which may be of interest for sample size calculation, as the Spanish sample, although small, is closer to the InProS population than the sample in the study of Gouze et al.
Variables measurement
2.1. The main outcome tool is a well described, well-known instrument to assess sensory processing. However, there is not enough information regarding measurement of sociodemographic, lifestyle and health-related variables. As most of them are measured using ad-hoc questionnaires, more information about some of these questionnaires should be provided, specifically regarding assessment of global physical activity, feeding problems and oral-facial problems, which are all complex variables.
For instance, do children complete some of the questionnaires, or do parents provide all the information? What aspects of feeding and oral-facial problems are examined? How are they assessed (i.e., on a Likert-point scale, on a yes/no response, etc.)?
2.2. If these ad-hoc questionnaires comprise several items, were their psychometric characteristics tested (i.e., internal consistency)? In limitations, authors say that “all the questionnaires used to gather information from the study participants were valid and reliable instruments used in previous research”. It would be necessary to provide data about their validity and reliability or to explicitly refer to studies that previously used the questionnaires and where readers can look for further information.
2.3. If I have understood correctly, children will be classified as having a typical or an atypical sensory profile using a cutoff of 155 in the Short Sensory Profile. A total score below 155 in the Short Sensory Profile corresponds to having both probable and definite differences. However, it is recommended to establish presence of SPD using the definite difference cutoff (141 or below). Moreover, most of the prevalence rates mentioned throughout the article have been estimated with the definite difference and not the probable difference cutoff, and sample calculation was operated on this assumption. Authors could address this issue including a second analysis to explore differences between children with and without SPD, not only between children with typical and atypical sensory profile.
Author Response
Please, see the document attached.

Reviewer 2 Report
This research paper completely lacks of result section. I think the authors wanted to show their proposal or future plan, it is not suitable for a research publication.
Indeed, Discussion section also is not related to their findings.
Author Response
Please, see the document attached with the responses.

Reviewer 3 Report
Thank you for the opportunity to review this paper. This study will add to the body of knowledge that already exists on the subject of Sensory Processing and is to be welcomed. Under procedure for enrolment perhaps the authors could make explicit that children with disabilities were not included in this study and that no children with disabilities attend these 21 schools? I assume that this was a criteria for inclusion/non-inclusion of participants?
Author Response

(The authors gave the same response as above.)

Round 2
Reviewer 1 Report
I would like to thank the authors for addressing all the points. I think this paper offers relevant and valuable information and will be a nice contribution to IJERPH.
Author Response
We thank the reviewer for his/her positive consideration of our manuscript and valuable comments that have been of great help to improve considerably the article.

Reviewer 2 Report
Authors failed to give an appropriate answer to my comments.
Even if this study could be shifted from article to protocol, it always regards a future plan, not an experimental protocol that could be published.
Author Response
We want to apologize to the reviewer for not having addressed appropriately to his/her query. The present article describes the research protocol of the InProS project, a cross-sectional study of parental and children’s sociodemographic and lifestyle features and children’s sensory processing. It provides the general outline of the study including the rationale, procedures, plan analysis, primary and secondary outcomes and dissemination of the research results. This type of articles may enhance transparency of research, reducing publication bias and also preventing selective publication and selective reporting of research outcomes, among other advantages. In this sense, these articles are of great interest to scientific journals. In our article, we have followed a similar structure used in other study protocols published in the IJERPH (please, see for more details: https://www.mdpi.com/1660-4601/16/9/1496; https://www.mdpi.com/1660-4601/16/19/3548). Modestly, we think that our article is suitable for the IJERPH and we do not understand the reviewer’s comment.
As indicated, this is a cross-sectional study and the data collection was made from February to May 2016. For this reason, this is not a study in an experimental stage. Regarding the future plan, the findings from the study will be presented at international meetings and will be published in open access peer reviewed journals. Furthermore, this study will be the subject of a doctoral thesis (P.F.-P.) and will meet the requirements the PhD program at the Miguel Hernández University (Alicante, Spain).
